# Dual Chamber Pacemaker Implant in Coronary Sinus Leading to Several Complications

**DOI:** 10.3390/diagnostics14222465

**Published:** 2024-11-05

**Authors:** Nancy Wassef, Mina Ibrahim, Christine Botrous, Amr Anos, Kai Hogrefe, Janaka Pathiraja

**Affiliations:** 1Gloucestershire Hospitals Foundation Trust, Cheltenham General Hospital, Sandford Road, Cheltenham GL53 7AN, UK; 2Bristol Royal Infirmary, Upper Maudlin Street, Bristol BS2 8HW, UK; 3Kettering General Hospitals, Rothwell Road, Kettering NN16 8UZ, UK

**Keywords:** heart block, permanent pacemaker, complications, learning points

## Abstract

Permanent pacemaker implantation is a low-risk procedure. However, complications may occur at a rate of around 4–8%. We present a case where initial implantation resulted in complications that could have been avoided by meticulous assessment of lead position in different projections and early post-procedure X-ray that would have delineated other serious complications. We present a case where the right ventricular lead was placed in the coronary sinus, which resulted in the loss of pacing capture with further syncope after the pacemaker implant. This was apparent in the post-procedure electrocardiogram (ECG) with right bundle branch pacing and the lead was repositioned in the right ventricular apex the following day. Furthermore, the patient was discharged home without a chest X-ray (CXR), and she represented a week later with a haemo-pneumothorax and pericardial effusion. A chest drain was placed and was discharged after a slow recovery following several complications that could have been avoidable.

An eighty-seven-year-old female patient who has no co-morbidities and is not on any medications presented with syncope and a documented high-grade atrioventricular block (Figure 1). She had a dual chamber pacemaker implant with difficulty in placing the right ventricular (RV) lead, which was arduous, and multiple positions were checked until a final position was accepted. Afterward, she had another episode of syncope with loss of pacing capture in the cardiac care unit (Figure 2A–C). This was confirmed by pacing checks with loss of capture at the highest output.

The patient had repositing of the RV lead in the right ventricular apex, and the following ECG showed left bundle branch block (LBBB) pattern pacing (Figure 3). Pacing checks were satisfactory after repositioning, and she was discharged home without a CXR.

A week later, she was re-admitted with worsening dyspnoea. CXR and CT revealed haemopericardium and haemothorax, and she was treated with a chest drain and conservatively managed for pericardial effusion (Figure 4 and Figure 5). She had a slow recovery and was discharged home after treatment with no further symptoms during follow-up in the pacing clinic.


**Learning Points**


Pacemaker complications may occur at a rate of around 4–8% [1].

ECG evaluation post-pacemaker implantation is essential, and pseudo RBBB with late V4 transition should raise the suspicion of coronary sinus pacing of the right ventricular lead [2].

ST elevation with injury current is an essential step during implant that alerts the operator to extracardiac placement and post-procedure complications [3].

LAO (Left anterior Oblique) projection will allow the operator to identify right ventricular versus coronary sinus lead placement during the procedure [2,3,4].

Chest X-ray and pacing interrogation are essential after all pacemaker implantations [5].


**Conclusions**


This case revealed several valuable learning points that are highly educational. The puncture site caused the pneumothorax, which would have been noticed by CXR had it been performed before discharge. We suggest a simple CXR protocol to all patients, even if the cephalic approach was accessed, which will identify lead positions, displacements, or loose connections to the pacemaker box [5].

In addition, we believe that the first positioning of the right ventricular lead led to perforation into the right ventricle, which caused pericardial effusion. This was noticed with pacing checks with ST depression injury pattern rather than elevation, so it was repositioned but, unfortunately, was placed in the middle cardiac vein. Adopting a post-procedure quick bedside echocardiography after any patient with ST depression injury pattern during implant will unveil effusion and prevent mortality with tamponade [3]. This will identify patients who can be discharged early versus those who need longer monitoring.

Furthermore, had the last position of the right ventricular lead been checked in LAO projection, it would have shown the position in the middle cardiac vein of the coronary sinus and would have been repositioned rather than accepting an unstable position with lead movement and loss of capture, resulting in further syncope and the need for another procedure. We promote the importance of the LAO approach to identify the true position of the right ventricular lead, identifying the coronary sinus position if the right ventricular lead crosses the midline [2,4].

## Figures and Tables

**Figure 1 diagnostics-14-02465-f001:**
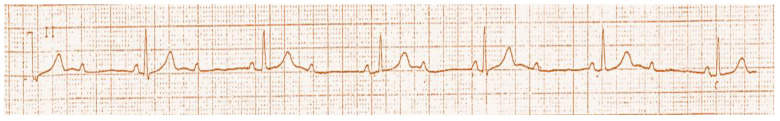
Demonstration of the Mobitz type two, second-degree atrioventricular block.

**Figure 2 diagnostics-14-02465-f002:**
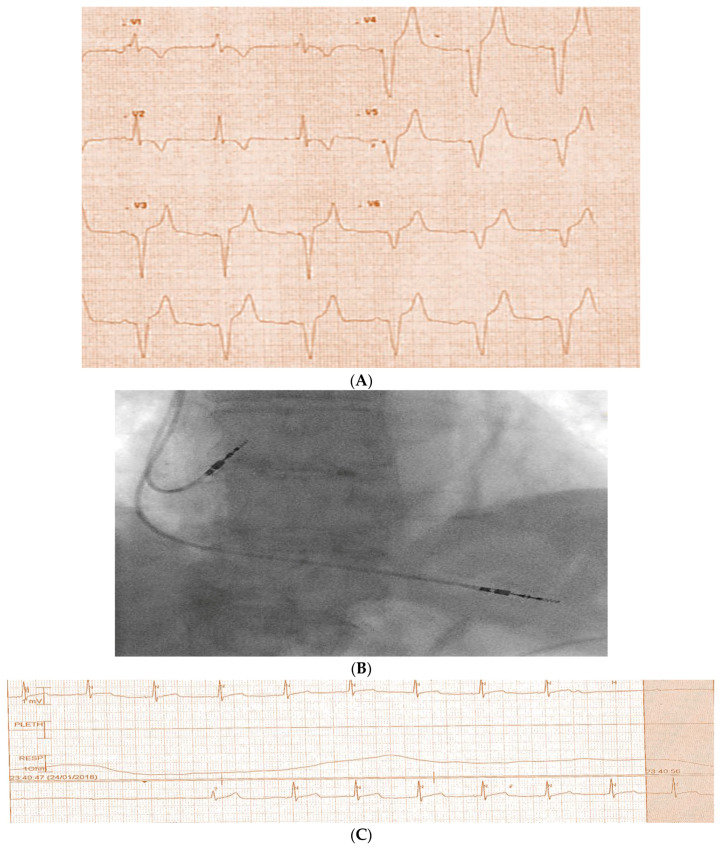
(**A**) Demonstrates ECG-paced ventricular rhythm with right bundle branch block (RBBB) pattern. (**B**) Demonstrates dual chamber final implant by fluoroscopy, with right ventricular lead in the middle cardiac vein position rather than the right ventricular apex. (**C**) Telemetry monitor revealing sinus rhythm with a long pause after pacemaker implant.

**Figure 3 diagnostics-14-02465-f003:**
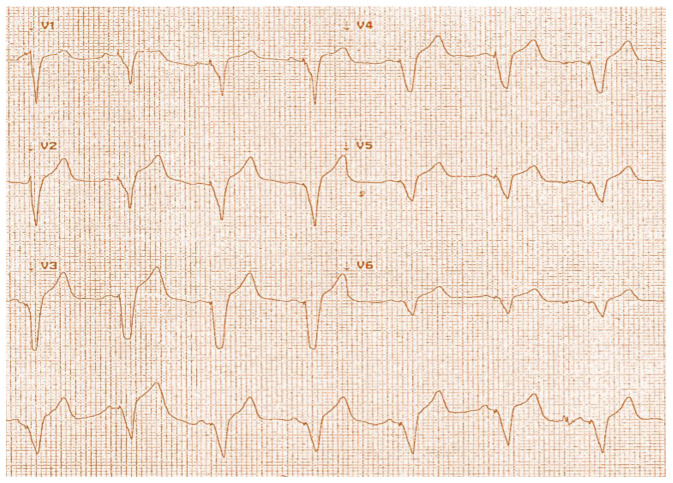
ECG showing LBBB pattern pacing.

**Figure 4 diagnostics-14-02465-f004:**
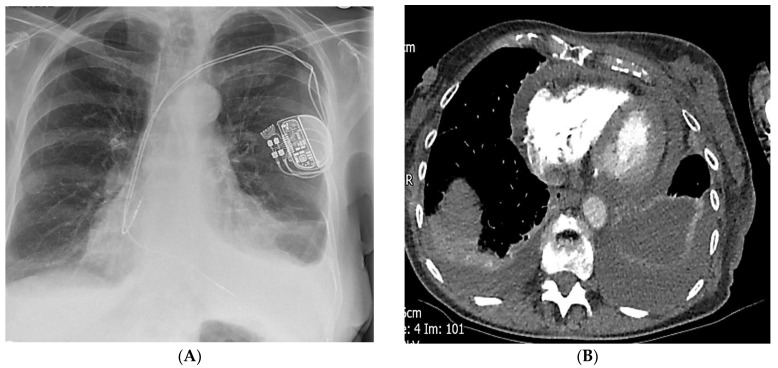
(**A**) CXR demonstrates haemo-pneumothorax with dual chamber pacemaker leads in position. (**B**) CT showing bilateral pleural and pericardial effusion. (**C**) Echocardiography from subcoastal and apical windows showing moderate pericardial effusion, with inferior vena cava (IVC) collapse of more than 50%.

**Figure 5 diagnostics-14-02465-f005:**
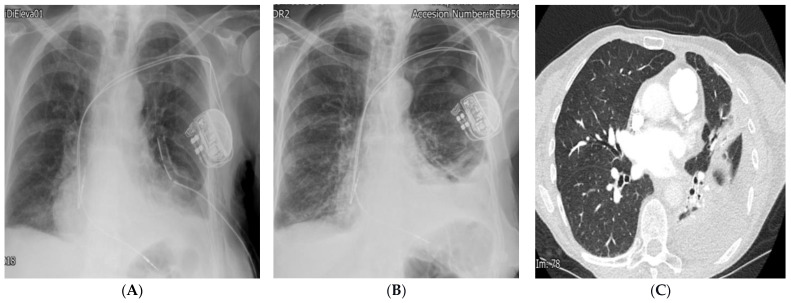
(**A**) CXR demonstrates a left-sided chest drain. (**B**) CXR demonstrates residual left haemo-pneumothorax after drain removal. (**C**) CT revealing residual loculated haemothorax.

## Data Availability

The data are available on request from the authors.

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
