# Peer review of "Dual Chamber Pacemaker Implant in Coronary Sinus Leading to Several Complications"

_diagnostics, 2024, doi:10.3390/diagnostics14222465_

Round 1
Reviewer 1 Report
Comments and Suggestions for Authors
The case is well-presented, though it could be enhanced by including more information on the patient's medical history, such as any comorbidities or previous cardiac conditions, to provide a more comprehensive clinical context.
In the discussion, the authors note that the complications might have been avoided with more careful assessment. To improve this section, they could propose specific protocols or checklists to be used during pacemaker implantation, which would help prevent such complications. This would introduce a practical element and increase the discussion's applicability to clinical practice. Authors should stress the easyness that we’ve reached nowadays in the management of leads complications, included lead extraction (Safety and feasibility of same-day discharge following uncomplicated transvenous lead extraction. J Cardiovasc Electrophysiol. 2024 Feb;35(2):278-287. doi: 10.1111/jce.16147.)
The conclusion could be strengthened by offering a clearer summary of the key learning points and their practical implications for clinical settings. While the learning points are currently well outlined, restating their significance in the conclusion would help solidify the message for the readers.
In certain parts of the manuscript, such as the figure captions (e.g., Figure 2), the wording can be simplified for better clarity.
Author Response
Reviewer One:
-The case is well-presented, though it could be enhanced by including more information on the patient's medical history, such as any comorbidities or previous cardiac conditions, to provide a more comprehensive clinical context. à
The patient had no co-morbidities and was always fit and well, not on any medications before presenting with heart block and we have added this to history.
-In the discussion, the authors note that the complications might have been avoided with more careful assessment. To improve this section, they could propose specific protocols or checklists to be used during pacemaker implantation, which would help prevent such complications. This would introduce a practical element and increase the discussion's applicability to clinical practice. Authors should stress the easyness that we’ve reached nowadays in the management of leads complications, included lead extraction (Safety and feasibility of same-day discharge following uncomplicated transvenous lead extraction. J Cardiovasc Electrophysiol. 2024 Feb;35(2):278-287. doi: 10.1111/jce.16147.) The conclusion could be strengthened by offering a clearer summary of the key learning points and their practical implications for clinical settings. While the learning points are currently well outlined, restating their significance in the conclusion would help solidify the message for the readers. à
The conclusion was added, and we are extremely grateful for this suggestion as it highlights the importance of our learning points.
We have added suggestions of specific points that will help to delineate complications that we have encountered in our case. We felt that adding the conclusion that clarified this specific case learning points, was important which we felt it allows us to convey clear messages to the reader.
-In certain parts of the manuscript, such as the figure captions (e.g., Figure 2), the wording can be simplified for better clarityà The figure caption wording was simplified, in addition image quality was improved significantly.
Sincerely,
Nancy Wassef
Cardiology Consultant
Gloucestershire hospitals Foundation Trust
Reviewer 2 Report
Comments and Suggestions for Authors
The images with tha abstract are interesting enough for publication.
The first shown ECG is of low quality. If a better sample could be presented, it would enhance the case.
The Medical imaging is of sufficient quality and can be published as such,
The use of language is acceptable.
Author Response
Reviewer Two:
-The images with the abstract are interesting enough for publication.
-The first shown ECG is of low quality. If a better sample could be presented, it would enhance the case. Thank you, we agree as image quality when was reviewed in different PCs were not adequate enough which we addressed and enhanced the image quality for the figure 1 and the DPI was raised to 600 as requested, in addition we have added better ECG images with 600 DPI to other figures.
-The Medical imaging is of sufficient quality and can be published as such,
Sincerely,
Nancy Wassef
Cardiology Consultant
Gloucestershire hospitals Foundation Trust
Reviewer 3 Report
Comments and Suggestions for Authors
The authors report a case of pericardial effusion and pleural effusion occurring after permanent pacemaker (PPM) implantation, with a suggestion that the right ventricular (RV) lead might have been placed in the coronary sinus during the procedure.
I have several questions and suggestions regarding this report:
1. What does the author believe to be the cause of the pericardial and pleural effusions? Were the effusions tested for their composition? Were they purely hemopericardium and hemothorax, or were other causes considered? Please provide a biochemical analysis of the fluid drained from the pleural effusion.
2. Does the author think the pericardial and pleural effusions are related to malposition of the PPM RV lead? Specifically, does the author suspect that the first attempt at placing the lead in the coronary sinus might be responsible, or is there a possibility that after repositioning, the lead perforated the RV wall? Since a CT scan was performed, it should be possible to identify the exact position of the RV lead and clarify this issue.
3. As this is an "interesting image" topic, and the author believes the RV lead was initially placed in the coronary sinus, corresponding images should be provided to support this.
4. In summary, I find it difficult to discern the key message the authors wish to convey. What do they believe caused the patient's pericardial and pleural effusions? What is the primary lesson for the readers? If the scope of an “interesting image” article is too limited to present all the data and reasoning, the author might consider submitting this case as a full case report instead.
Author Response
Reviewer Three:
Thank you for the questions, which were important to address.
The authors report a case of pericardial effusion and pleural effusion occurring after permanent pacemaker (PPM) implantation, with a suggestion that the right ventricular (RV) lead might have been placed in the coronary sinus during the procedure. I have several questions and suggestions regarding this report:
- What does the author believe to be the cause of the pericardial and pleural effusions? Were the effusions tested for their composition? Were they purely hemopericardium and hemothorax, or were other causes considered? Please provide a biochemical analysis of the fluid drained from the pleural effusion. à
We believe that the cause of the effusion occured during the first implant of the RV lead, which was reported to be very difficult ( we added this to the history) as the patient with initial RV apex position had good capture but had ST depression injury pattern rather than elevation by pacing checks, which was noticed and repositioned by operator, but no echocardiography was done after. We believe this was the cause of the effusion.
It was pure Haemothorax, and all samples were sent and came negative with MCS, biochemistry and cytology.
- Does the author think the pericardial and pleural effusions are related to malposition of the PPM RV lead? Specifically, does the author suspect that the first attempt at placing the lead in the coronary sinus might be responsible, or is there a possibility that after repositioning, the lead perforated the RV wall? Since a CT scan was performed, it should be possible to identify the exact position of the RV lead and clarify this issue. à
We believe that the coronary sinus middle cardiac vein position was not stable but was not the cause of the effusion, but the effusion occurred before with the previous initial apical position which was the reason the operator changing the position several times when he saw the ST depression with pacing checks, which was not seen afterwards in other pacing checks, but the injury pattern was minimal with no clear ST elevation, but was accepted by the operator.
The CT was done later after re-admission a week later after she was discharged following the second procedure where the lead was repositioned into a final position. This CT did not show any lead puncturing through the RV apex, nor it showed any clear tear, which was likely a micro-tear that was already repositioned in the first procedure. Lastly, the subcostal echo image is showing the lead position within the RV with no puncturing through (Figure 4C).
- As this is an "interesting image" topic, and the author believes the RV lead was initially placed in the coronary sinus, corresponding images should be provided to support thisà The fluoroscopy image has been included (Figure 2B) , as explained in our case the failure of operator to check in LAO view which would have identified the CS MCV position, which we all use when placing a CRT to assess the vein we select. The ECG shows the pacing spike in V1 with the fluoroscopy image and in addition to the blunt injury pattern, which was accepted by operator with no clear ST elevation, all were identifying MCV CS position which we referenced in reference 2, which was very educative to us.
- In summary, I find it difficult to discern the key message the authors wish to convey. What do they believe caused the patient's pericardial and pleural effusions? What is the primary lesson for the readers? If the scope of an “interesting image” article is too limited to present all the data and reasoning, the author might consider submitting this case as a full case report instead. à
We have added a full new conclusion paragraph to explain the learning outcome for the readers. This has made our learning points clearer and we wish to thank you for pointing this out, as it reads better than before.
We have initially submitted as a case report, but as this is a special edition to Diagnostics Journal, we were advised in an email from the Journal Editor to change it to image in cardiology, which we have happily done and changed it from the original case report to Image in cardiology to fit the special edition for pacing to the journal as requested, as we were advised there will be no cases in this journal edition but images are accepted.
Round 2
Reviewer 1 Report
Comments and Suggestions for Authors
the authors have approriately answered to all of my comments.
Reviewer 2 Report
Comments and Suggestions for Authors
The improvements are sufficient to merit publication.
Reviewer 3 Report
Comments and Suggestions for Authors
The author provided reasonable explanations for the questions raised, and I appreciate their effort. However, there is still room for improvement in revising the article. If not for the author's responses, it would be challenging to fully understand this case based solely on the article.
The author should move some content from the conclusion to the history section, as the current description of the history is too brief. This brevity makes the explanations in the conclusion appear abrupt to readers. For example, after reading the history, I cannot discern that the hemothorax and pericardial effusion were caused by different reasons. Additionally, the history does not clarify which step led to the pericardial effusion.
The author should reorganize the article to make it a more comprehensible case report.
Lastly, I still recommend that, since the submission is under the 'Interesting Image' category, the provided images should indeed be interesting. The current images do not meet this criterion. There are no CXR or echocardiography images from before and after the pacemaker placements to demonstrate the timeline or cause of the pericardial effusion and hemopneumothorax. Additionally, there are no LAO views during the procedure to confirm the misplacement of the RV lead into the coronary sinus. This article might be more suitable for a journal that accepts such case reports.